# Fixation-pattern similarity analysis reveals adaptive changes in face-viewing strategies following aversive learning

Lea Kampermann[1], Niklas Wilming[2], Arjen Alink[1], Christian Büchel[1], Selim Onat[1]*

[1]Department of Systems Neuroscience, University Medical Center Hamburg-Eppendorf, Hamburg, Germany; [2]Department of Neurophysiology and Pathophysiology, University Medical Center Hamburg-Eppendorf, Hamburg, Germany

**Abstract** Animals can effortlessly adapt their behavior by generalizing from past aversive experiences, allowing to avoid harm in novel situations. We studied how visual information was sampled by eye-movements during this process called fear generalization, using faces organized along a circular two-dimensional perceptual continuum. During learning, one face was conditioned to predict a harmful event, whereas the most dissimilar face stayed neutral. This introduced an adversity gradient along one specific dimension, while the other, unspecific dimension was defined solely by perceptual similarity. Aversive learning changed scanning patterns selectively along the adversity-related dimension, but not the orthogonal dimension. This effect was mainly located within the eye region of faces. Our results provide evidence for adaptive changes in viewing strategies of faces following aversive learning. This is compatible with the view that these changes serve to sample information in a way that allows discriminating between safe and adverse for a better threat prediction.

DOI: https://doi.org/10.7554/eLife.44111.001

*For correspondence:
selim.onat@free-now.com

Competing interest: See
page 20

Reviewing editor: Thorsten
Kahnt, Northwestern University
Feinberg School of Medicine,
United States

## Introduction

To avoid costly situations, individuals must be able to rapidly predict future adversity based on previously learnt aversive associations, as well as actively sampled information from the environment. However, new situations are seldom the same as previously experienced ones. To judge whether a situation holds potential harm, a careful balance between stimulus generalization and selectivity is needed. While fear generalization makes it possible to promptly deploy defensive behavior when similar situations are encountered anew (*Guttman and Kalish, 1956*; *Hovland, 1937*; *Pavlov, 1927*; *Shepard, 1987*; *Spence, 1937*; *Struyf et al., 2015*; *Tenenbaum and Griffiths, 2001*), selectivity ensures that only truly aversive stimuli are recognized as aversive (*Li et al., 2008*; *Onat and Büchel, 2015*), thus avoiding costly false alarms. In real-world situations adversity predictions are based on sensory samples collected through active exploration (*Henderson, 2003*; *Itti and Koch, 2001*). A central part of active exploration are eye-movements which can rapidly determine what information is available in a scene for recognizing adversity (*Dowd et al., 2016*). Yet, it is not known in how far representations of adversity interact with active exploration during viewing of complex visual information. Here we investigated this question by comparing exploration strategies during viewing of faces before and after aversive learning.

The approach to study exploration of faces is motivated by the fact that faces are characterized by subtle idiosyncratic differences on the stimulus continuum. Humans explore such complex stimuli serially (*Itti and Koch, 2001*; *Onat et al., 2014*; *Parkhurst et al., 2002*; *Tatler et al., 2005*) and rely heavily on their overt attentional resources due to well documented perceptual bottleneck for

forming detailed representations of complex scenes (*Ahissar and Hochstein, 2004*; *Hochstein and Ahissar, 2002*; *Treisman and Gelade, 1980*). Therefore, eye-movement recordings can mirror strategies used for foraging information from different faces, which in turn informs us on the way how aversive learning changes the exploration of relevant stimuli (*Dowd et al., 2016*; *Henderson and Hayes, 2018*; *König et al., 2016*).

This way, face viewing behavior offers an ideal test bed for investigating changes in active exploration strategies through learning. First, active viewing of faces is a key ability during daily social interactions (*Cerf et al., 2009*; *End and Gamer, 2017*) where detecting minute differences in the configuration of facial elements is crucial for inferring the identity or emotional content of a face (*Adolphs, 2008*; *Jack et al., 2014*; *Peterson and Eckstein, 2012*). For humans, it is therefore a natural choice of stimuli to investigate how exploration strategies change with learning. Second, the universal spatial configuration of facial elements makes it easily possible to generate faces with subtle differences that globally form a perceptual similarity continuum (*Dunsmoor et al., 2011*; *Onat and Büchel, 2015*). These key features make it possible to use a task that mimics a real-world exploration context, and therefore offers the possibility to probe changes in exploration strategies with aversive learning along a parametrically controlled stimulus continuum.

We investigated fear generalization using a two-dimensional perceptual space (*Figure 1A*, *Figure 1—figure supplement 1*) with faces arranged along a circle within this space (*Butter, 1963*; *Onat and Büchel, 2015*). By pairing one item with an aversive outcome and keeping the most dissimilar one neutral (opposite face separated by 180 degrees), we introduced an adversity gradient defined exclusively along one perceptual dimension. The perceptual space can therefore be decomposed into threat-*specific* and *unspecific* components (*Figure 1B*, middle panel), where the latter models perceptual similarity independent of adversity. This set of stimuli made it possible to dissociate independent contributions of perceptual factors related to similarity as such, from those relevant for the prediction of threat.

We based our main analysis on multivariate eye-movement patterns and introduced a similarity-based multivariate method that we termed fixation-pattern similarity analysis (FPSA). Instead of focusing on how often arbitrary parts of a face are fixated (*Hessels et al., 2016*; *Malcolm et al., 2008*; *Schurgin et al., 2014*) as in traditional fixation count-based approaches, FPSA uses simultaneously all available fixations to derive a similarity metric, similar to representation similarity analysis in brain imaging (*Kriegeskorte, 2008*). This way, it respects the strong individuality of eye-movements, that is idiosyncrasy (*Coutrot et al., 2016*; *Kanan et al., 2015*; *Mehoudar et al., 2014*; *Walker-Smith et al., 1977*). Another benefit of this approach is that the circular organization of our stimuli allowed us to formulate hypotheses on how the similarity relationships between exploration patterns could change: More explicitly, we could model changes along the adversity-specific direction, while controlling for changes along the adversity independent direction defined purely by perceptual differences. This way, the latter, unspecific dimension served as within-subject control condition for changes independent from aversive learning. We hypothesized three possible scenarios on how aversive learning might change exploration along the two-dimensional stimulus space (*Figure 1B–E*, middle panels).

First, the *Perceptual Expansion* hypothesis predicts that exploration patterns should enable evaluation of perceptual similarity with the CS+ face. This would result in exploration strategies that strongly mirror the physical similarity relationships between faces (*Figure 1C*) and lead potentially to globally increased dissimilarity following learning. As similarity information varies both on the specific as well as the unspecific dimensions, the *Perceptual Expansion* hypothesis predicts a stronger, but more importantly equal contribution of the underlying specific and unspecific components (*Figure 1C*, middle panel).

Second, it is possible that aversive learning induces changes in eye-movements specifically along the adversity specific dimension, in a way that exploration strategies would be tailored to target locations that are discriminative of the CS+ and CS− faces. This would lead to exploration patterns becoming more similar for faces sharing similar features with the CS+ and CS− faces, while simultaneously predicting an increased dissimilarity between these two sets of exploration patterns (*Figure 1D*). Increased similarity only along the adversity-relevant dimension would then result in an ellipsoid representation of similarity relationships. Therefore, the *Adversity Gradient hypothesis* would lead to an increase of the adversity specific component without influencing the unspecific component (*Figure 1D*, middle panel).

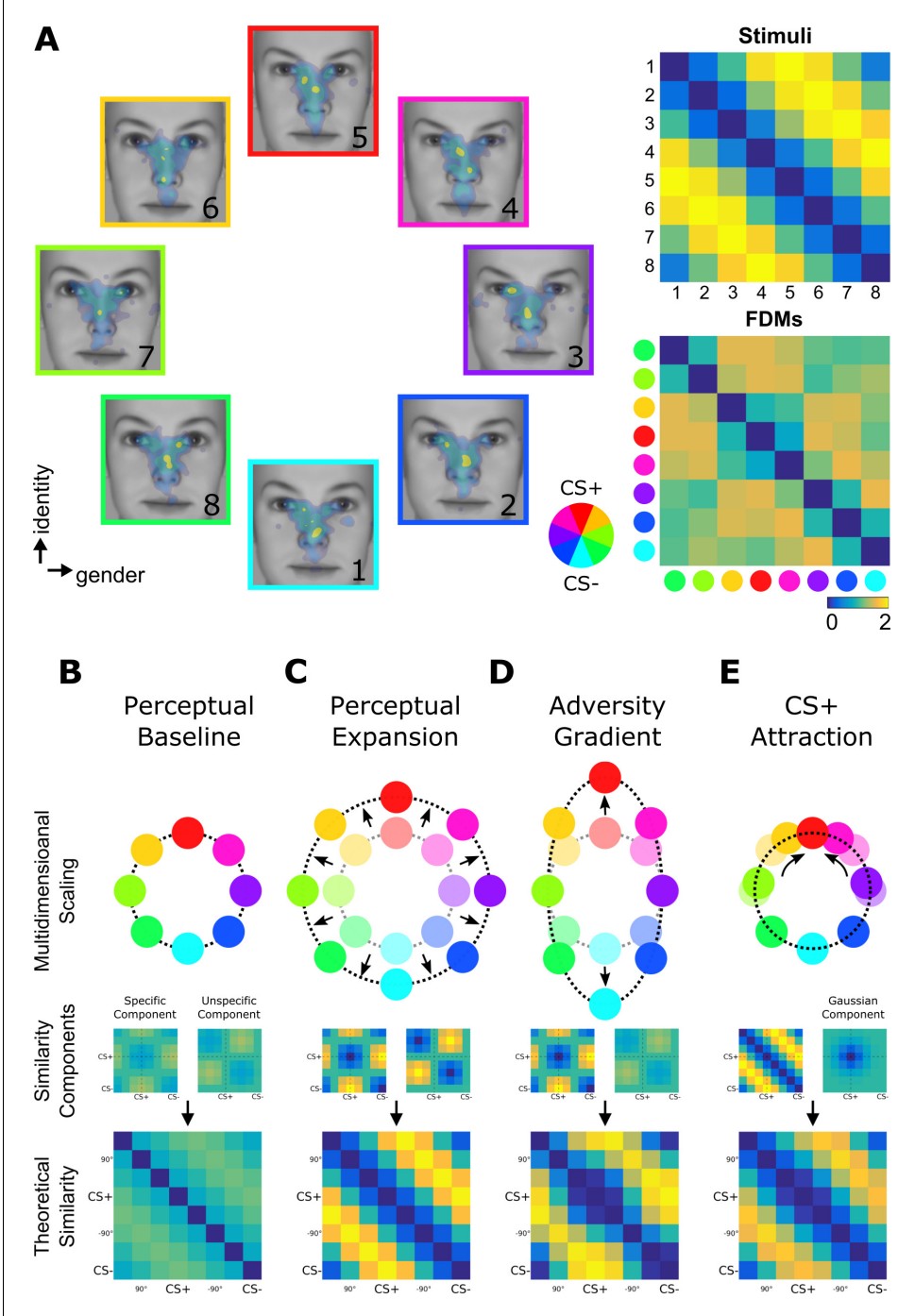

**Figure 1.** Fixation-pattern similarity analysis. (**A**) eight exploration patterns (colored frames) from a representative individual overlaid on eight face stimuli (1 to 8) calibrated to span a circular similarity continuum across gender and identity dimensions. A pair of maximally dissimilar faces was randomly selected as CS+ (red) and CS– (cyan; see color wheel). Similarity between the eight faces was calibrated to have a perfect circular organization with lowest dissimilarity (blue) between neighbors, and highest dissimilarity (yellow) for opposing pairs. FPSA summarizes the similarity relationship between the eight exploration patterns as a symmetric 8 × 8 matrix (bottom right panel). 4th and 8th columns (and rows) are aligned with the CS+ and CS–, respectively. (**B–E**) Multidimensional scaling representation of four theoretical similarity relationships investigated with FPSA (top row). Each colored node represents one exploration pattern (red: CS+; cyan: CS–), where internode distances are proportional their dissimilarity (bottom row). Shaded nodes in (**C–E**) depict the pre-learning state in (**B**). Dissimilarity matrices are further decomposed onto basic similarity components (middle row) centered either on

*Figure 1 continued on next page*

*Figure 1 continued*

the CS+/CS– (specific component) or +90°/–90° faces (unspecific component). A third component shown in (E) is uniquely centered on the CS+ face (*Gaussian* component). In (B), equal contribution of basic components results in circularly similar exploration patterns. In (C), a stronger equal contribution results in a better global separation of all exploration patterns, that is expansion (denoted by radial arrows). In (D), a stronger contribution of the specific component results in a biased separation of exploration patterns specifically along the adversity gradient defined between the CS+ and CS– nodes. In (E), the Gaussian component centered on the CS+ face can specifically decrease the dissimilarity of exploration patterns for faces similar to the CS+, resulting in circularly shifted nodes (circular arrows) while preserving the global circularity of the similarity relationships.
DOI: https://doi.org/10.7554/eLife.44111.002

The following figure supplements are available for figure 1:

**Figure supplement 1.** Face stimuli.
DOI: https://doi.org/10.7554/eLife.44111.003
**Figure supplement 2.** Calibration of faces using a simple V1 model tuned to human psychophysics.
DOI: https://doi.org/10.7554/eLife.44111.004

---

Third, exploration strategies could change so that viewing patterns are tailored to quickly identify the CS+. A new sensorimotor strategy exclusively for the adversity predicting face would lead to a localized change in the similarity relationships around the adversity-predicting CS+ face (*Figure 1E*). Thus, the hypothesis of *CS+ Attraction* predicts an increased similarity of exploration patterns for faces closely neighboring the CS+ face, decaying proportionally with increasing dissimilarity to the CS+ face. This strategy is not exclusive and does not directly map to the specific or unspecific components.

In sum, using FPSA we analyzed the similarity relationships between exploration patterns during viewing of faces. We provide first evidence that exploration patterns during viewing of faces can be adaptively tailored during generalization following aversive learning. First, aversive learning changed exploration patterns in subtle ways that were not captured by univariate fixation counts, for example in predefined regions of interest. Second, before learning, exploration patterns showed an approximately circular similarity structure that followed the physical stimulus similarity structure. Third, after learning the similarity structure changed specifically along the adversity gradient, indicating that CS + and CS– exploration patterns were modified, while the similarity between other faces remained largely unchanged.

## Results

We created eight face stimuli that were organized along a circular similarity continuum characterized by subtle physical differences in facial elements across two dimensions (gender and identity; see *Figure 1—figure supplement 1* for stimuli). We calibrated the degree of similarity between faces using a simple model of the primary visual cortex known to mirror human similarity judgments (*Yue et al., 2012*) (see *Figure 1—figure supplement 2* for calibration). The physical similarity relationship between all pair-wise faces conformed with a circular organization (*Figure 1A*, top right panel), such that dissimilarity varied with angular difference between faces (lowest for left and right neighbors and highest for opposing faces) with equidistant angular steps. Participants (n = 74) freely viewed these faces before and after an aversive learning procedure (*Figure 2A*) while we measured their eye-movements. During the conditioning phase, one of the eight faces was introduced as the CS+, being partially reinforced with an aversive outcome (UCS, mild electric shock in ~30% trials). The CS– was the face most dissimilar to the CS+ (separated by 180°) and was not reinforced. During the subsequent generalization phase, all faces were presented again and the CS+ continued to be partially reinforced to prevent extinction of the previously learnt association. These reinforced trials were excluded from the analysis. To ensure comparable arousal states between the baseline and generalization phases, we administered UCSs also during the baseline period, however they were fully predictable as their occurrence was indicated by a shock symbol (*Figure 2A*). Furthermore, we inserted null trials during all phases (i.e. trials without face presentation but otherwise exactly the same) in order to obtain reliable baseline levels for skin-conductance responses.

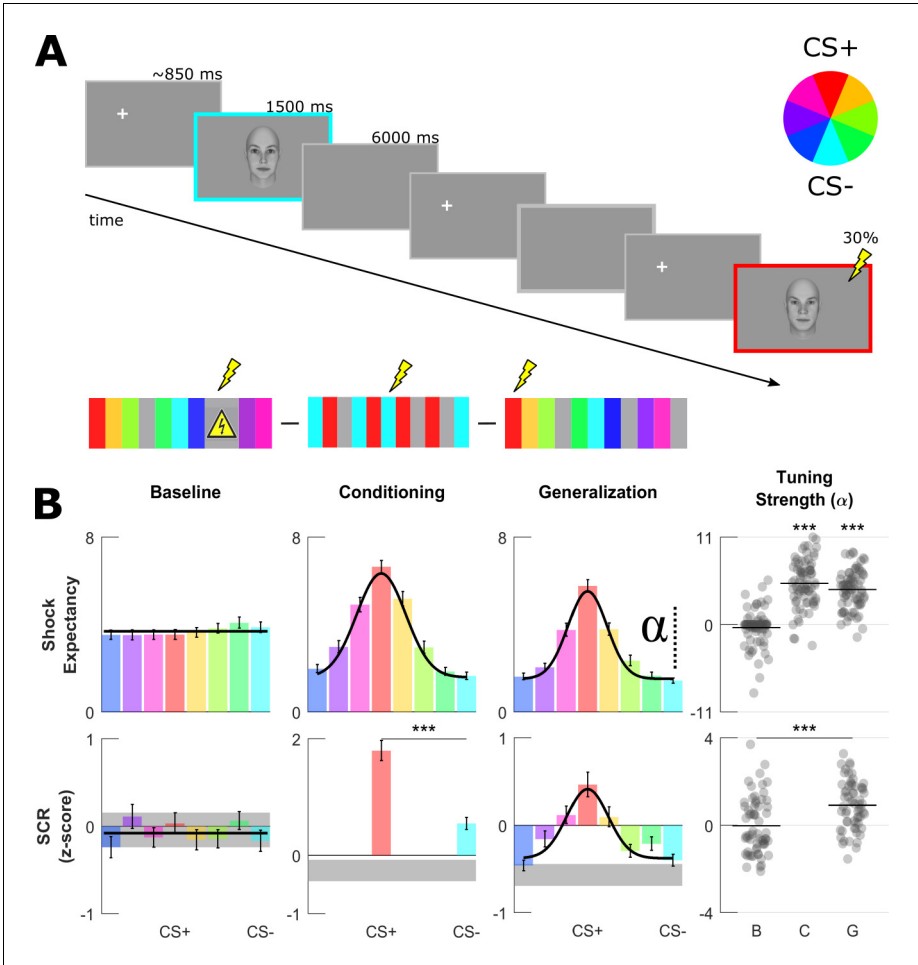

**Figure 2.** Aversive learning procedure and univariate generalization profiles. (**A**) On every trial, one randomly selected face was presented for 1.5 s preceded by a fixation cross placed outside of the face on either the left or right side. In null trials, no face was shown (*gray frame*) resulting in a SOA of ~6 or~12 s. For each volunteer, a pair of most dissimilar faces was randomly selected as the CS+ (*red*) and CS– (*cyan*, see color wheel). During baseline, UCSs (shock sign) were completely predictable by the presentation of a triangular signboard. During conditioning and generalization, the CS+ face was paired with an aversive outcome in ~30% of trials allowing recording of responses from non-reinforced CS+ trials. (**B**) Group-level fear-tuning based on subjective ratings of UCS expectancy (n = 74) and SCR (n = 63) for different phases. Responses are aligned to the CS+ of each volunteer separately (errorbars: SEM across subjects). Black horizontal lines or curves indicate the winning model (p<0.001, log-likelihood ratio test), that is horizontal null model or the Gaussian model. Gray shaded areas in SCR depict response amplitudes evoked by null trials (mean and 95% CI). Scatter plots show amplitude parameter of Gaussian fits (denoted by alpha symbol) for each volunteer. Horizontal lines within the scatterplots depict group-level means, asterisks indicate significant differences in α (compared to baseline phase, paired t-test, ***: p<0.001).
DOI: https://doi.org/10.7554/eLife.44111.005

## Fear tuning profiles in subjective ratings and autonomic activity

As expected, the effect of learning was mirrored both in autonomous nervous system activity as well as subjective ratings of UCS expectancy (*Figure 2B*). During the conditioning phase, skin-conductance responses (SCR) were on average 3.8 times higher for CS+ than CS– (*Figure 2B*, middle panel bottom row, paired t-test, p<0.001) indicating that the CS+ face gained a stronger aversive quality already during the conditioning phase. In line with this view, expectancy ratings gathered right at the end of the conditioning phase were also highest for the CS+ face (*Figure 2B*, middle panel top row).

As in previous studies, we characterized fear generalization by computing fear tuning profiles based on subjective ratings and SCR. In both recording modalities, responses decayed with increasing dissimilarity to the CS+ face and reached minimal values for CS–. We modeled these with a Gaussian function centered on the CS+ face. At the group-level, model comparison favored the flat null model over the Gaussian in both recording modalities before learning (p=0.44 for SCR, p=0.17 for ratings, log-likelihood ratio test; black horizontal lines in *Figure 2B*). However, following the conditioning phase the Gaussian model fitted the data significantly better (comparison to flat null model, p<0.001 for SCR and subjective ratings, log-likelihood ratio test). Fear-tuning profiles at the single-subject level were in agreement with the overall group-level picture. As Gaussian generalization curves on the group level can equally emerge from binary generalization profiles, for example a boxcar shape spanning the CS+ and neighbors of different degrees, we tested whether a Gaussian would explain single-subject tunings better than a binary profile (optimized with regards to amplitude, width and offset). The Gaussian explained both ratings and SCR profiles of single subjects better than the binary function ($AIC_{Gauss} < AIC_{Bin}$, Ratings: 72/74 and 68/74 subjects in conditioning (median $\Delta AIC = -3.6$) and generalization phase (median $\Delta AIC = -3.6$), SCR: 58/63 subjects in the generalization phase (median $\Delta AIC = -3.9$). Therefore, we summarized fear-tuning profiles of individual participants with the amplitude parameter of the fitted Gaussian function, which characterizes the modulation depth of fear tuning (i.e. the strength of fear tuning) after accounting for baseline shifts. The average amplitude parameter following the conditioning phase was significantly bigger than the baseline phase in both recording modalities (paired t-test, p<0.001, see *Figure 2B*). In summary, univariate fear-tuning in SCR and subjective ratings confirmed that aversive learning was successfully established and transferred towards other perceptually similar stimuli.

## Multivariate fear tuning profiles in eye movements

We analyzed exploration behavior using fixation density maps (FDMs). These are two-dimensional histograms of fixation counts across space, smoothed and normalized to have unit sum (see Materials and methods section for details). When computed separately for different faces, FDMs indicate how much different parts of a given face receive attentional resources. For each volunteer, we computed a dissimilarity matrix that summarized all pairwise comparisons of FDMs (using 1 - Pearson correlation as a pattern distance measure) and averaged these after separately aligning them to each volunteer's custom CS+ face (shown always at the 4th column and row in *Figure 3A*). Furthermore, in order to gather an intuitive understanding of learning-induced changes in the similarity geometry, we used multidimensional scaling (jointly computed on the $16 \times 16$ matrices, that is all pairwise combinations of 8 conditions from baseline and generalization phase). Multidimensional scaling (MDS) summarizes similarity matrices by transforming observed dissimilarities as closely as possible onto distances between different nodes (*Figure 3B*) representing different viewing patterns in 2D, therefore making it easily understandable at a descriptive level.

Already during the baseline period the dissimilarity matrix was highly structured (*Figure 3A*). In agreement with a circular similarity geometry and the MDS depiction, lowest dissimilarity values ($1.05 \pm 0.01$; M ± SEM) were found between FDMs of neighboring faces (i.e. first off-diagonal), whereas FDMs for opposing faces separated by 180° exhibited significantly higher dissimilarity values ($1.21 \pm 0.01$; paired t-test, $t(73) = 7.41$, p<0.001). Using the Perceptual Baseline Model (*Figure 1B*, 1st column), we investigated the contribution of physical characteristics of the stimulus set to the observed pre-learning dissimilarity structure. This model uses a theoretically circular similarity matrix (consisting of equally weighted sums of specific and unspecific components) as a linear predictor (*Figure 1B*, theoretical similarity matrix). This model performed significantly better compared to a null model consisting of a constant similarity for all pairwise FDMs comparisons (for Perceptual Model adjusted $r^2 = 0.09$; log-likelihood-ratio test for the alternative null model: $p<10^{-5}$; $BIC_{NullModel} = -96.1$, $BIC_{Perceptual} = -244.7$; see *Supplementary file 1A* for the results of model fitting). We additionally fitted the Perceptual Model for every volunteer separately (*Figure 3C*, first white bar). Model parameters at the aggregate level were significantly different from zero (*Figure 3C*; $w_{Circle} = 0.09 \pm 0.01$, M ± SEM; $t(73) = 7.99$, p<0.001, $BIC_{Perceptual} = -3.5 \pm 1.8$, M ± SEM) indicating that exploration strategies prior to learning mirrored the physical similarity structure of the stimulus set. This provides evidence that fixation selection strategies are, at least to some extent, guided by physical stimulus properties during viewing of neutral faces.

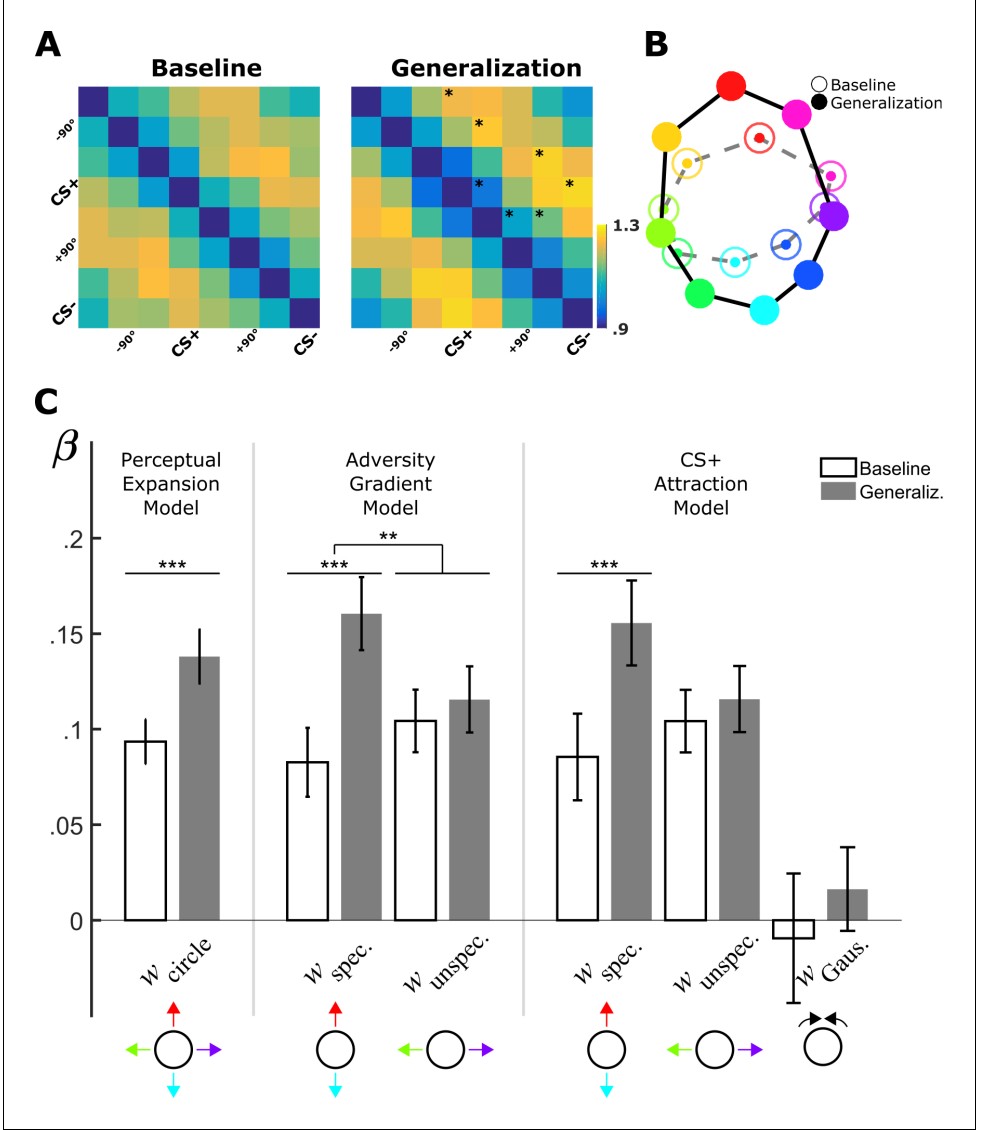

**Figure 3.** Fixation-pattern similarity analysis. (A) Dissimilarity matrices of exploration patterns for baseline (*left panel*) and generalization phases (*right panel*). Fourth and eight columns (and rows) are aligned with each volunteer's CS+ and CS– faces, respectively. Asterisks on the upper diagonal denote significant differences from the corresponding element in baseline. (B) Two-dimensional multidimensional scaling method used for visualization of the 16 × 16 dissimilarity matrix (not shown) comprising baseline and generalization phases. Distances between nodes are proportional to the dissimilarity between corresponding FDMs (*open circles*: baseline; *filled circles*: generalization phase; same color scheme as in *Figure 1*). (C) Bar plots (M ± SEM) depict predictor weights estimated for single-participants before (white bars) and after (gray bars) aversive learning (*Left:* Perceptual Expansion Model; *Middle:* Adversity Gradient Model; *Right:* CS+ Attraction Model). $w_{circle}$: weight for the circular component, which is the sum of equally weighted specific and unspecific components; $w_{specific}$/$w_{unspecific}$: weights for specific and unspecific components; $w_{Gauss}$: weight for Gaussian component centered uniquely on the CS+. (**: $p < 0.01$; ***: $p < 0.001$, paired *t-test*).

DOI: https://doi.org/10.7554/eLife.44111.006

The following figure supplements are available for figure 3:

**Figure supplement 1.** Correlation between FPSA anisotropy and tuning strength.
DOI: https://doi.org/10.7554/eLife.44111.007

**Figure supplement 2.** Model parameters for Adversity Gradient Model across subsequent test runs.
DOI: https://doi.org/10.7554/eLife.44111.008

**Figure supplement 3.** Gaussian Mixture Models on individual anisotropy effects.
DOI: https://doi.org/10.7554/eLife.44111.009

We observed significant changes between baseline and generalization dissimilarity values in an element-wise comparison (*Figure 3A*, indicated by asterisks). This provides evidence for learning-induced changes in the similarity relationships. Following learning, the circular Perceptual Model was again significant (adjusted $r^2$ = 0.35; p<0.001, log-likelihood ratio test), but now performed better compared to the baseline phase ($BIC_{Perceptual}$ = −244.7 for the baseline vs. $BIC_{Perceptual}$ = −1697.4 for the generalization phase; see *Supplementary file 1B* for model fitting results). Critically, we found a significant increase in the model parameter from baseline to generalization phase ($w_{Circle}$ = 0.13 ± 0.01; paired *t*-test, $t(73)$ = 4.03, p<0.001; *Figure 3C* compare two leftmost bars) suggesting a global increase in dissimilarity between FDMs. Overall, these results are compatible with the view that aversive learning led to a better separation of exploration patterns globally, in agreement with the Perceptual Expansion Model (*Figure 1C*).

However, as already visible in the multi-dimensional scaling method (*Figure 3B*), the separation between exploration patterns occurred mainly along the adversity gradient defined by the CS+ and CS− faces, whereas the separation along the orthogonal unspecific direction did not exhibit any noticeable changes. Regarding individual dissimilarity values, this effect was subtle, with an increase of average dissimilarity on the specific dimension from 1.21 ± 0.02 in baseline to 1.27 ± 0.02 in the generalization phase ($t(73)$ = 2.54, p<0.05, *paired t-test*), as compared to the orthogonal dimension (1.21 ± 0.03 vs. 1.20 ± 0.02, $t(73)$ = − 0.34, p = 0.73, *paired t-test*). Accordingly, in the generalization phase, dissimilarity along the specific dimension was bigger than along the unspecific dimension (1.27 ± 0.03 vs. 1.20 ± 0.02, $t(73)$ = 2.76, p<0.01, *paired t-test*). We thus extended the circular Perceptual Model to capture independent variance along the two orthogonal directions using the Adversity Gradient Model (*Figure 1D*). Model comparison indicated that the more flexible model performed better than the simpler Perceptual Expansion Model ($BIC_{Perceptual}$ = −1697.4 vs. $BIC_{Adversity}$. = −1957.5; adjusted $r^2$ = 0.48 vs. 0.35; see *Supplementary file 1C–D* for fitting results with the Adversity Gradient model on baseline and generalization phases, respectively). A two-factor (experimental phase ×predictor type) repeated measures ANOVA showed a significant main effect of experimental phase (i.e. before vs. after conditioning, $F(1, 73)$ = 6.29, p<0.001), as well as a significant interaction of predictor (specific vs. unspecific) with experimental phase $F(1,73)$ = 8.53, p<0.005). Accordingly, post-hoc analyses showed that while the unspecific component did not change from baseline to generalization phase ($t(73)$ = 0.75, p = 0.45), the specific component gained stronger contribution ($t(73)$ = 4.64, p<0.001), and these learning-induced changes were significantly larger in the specific as compared to unspecific component ($t(73)$ = 2.92, p<0.005, paired t-test). This observation provides evidence that increased overall dissimilarity was driven by changes in the scanning behavior specifically along the task-relevant adversity direction. We will refer to this difference in dissimilarity ($w_{specific}$ - $w_{unspecific}$) as *anisotropy*. To explore whether the effect was stable across individual subjects, we analyzed the prevalence of the effect, that is in how many subjects the anisotropy increased from baseline (base) to generalization (gen) phase ($w_{specific\_gen}$-$w_{unspecific\_gen}$ > $w_{specific\_base}$-$w_{unspecific\_base}$). 48 subjects out of 74 (65%) showed stronger anisotropy of scanning patterns after learning. Next, we studied whether explicit learning of shock contingencies predicted this increase in anisotropy and found that within the subjects whose shock expectancy ratings indicated successful conditioning (CS+>CS-, $n_{learner}$ = 61), the prevalence was slightly higher ($n_{anisotropy}$ = 42, i.e. 69%).

Given that only a portion of subjects showed the anisotropy effect, we tested whether this could be interpreted as evidence for model heterogeneity in the population. We thus employed a Gaussian mixture model analysis to fit the distribution of observed anisotropy values. We found that no Gaussian mixture model of more than one component outperformed the single component model (*Figure 3—figure supplement 3*). While the one and two component model were comparable in their BIC, the additional Gaussian component of the more complex model covered only a very small mixing proportion at the lower end of the distribution (4%, *Figure 3—figure supplement 3b*). Thus, we conclude that the present data does not provide enough evidence for the presence of multiple subpopulations.

The remodeling of the similarity geometry along the adversity gradient can also be accompanied by exploration strategies that are specifically tailored for the adversity predicting face but not for CS− resulting in localized changes only around the CS+ face. We subjected this view to model comparison by augmenting the previous model with a similarity component that consisted of a two-dimensional Gaussian centered on the CS+ face. Positive contribution of this predictor would lead

to more similar exploration patterns specifically around the CS+ (*Figure 1E*). It can thus capture changes in similarity relationships that are specific to the CS+ face. The model comparison procedure favored the simpler Adversity Gradient model over the augmented CS+ Attraction Model ($BIC_{Adversity.}$ = −1957.5 vs. $BIC_{CS+Attraction.}$ = −1923.6 during the generalization phase; adjusted $r^2$ = 0.48; see *Supplementary file 1E–F* for fitting results with CS+ Attraction Model in baseline and generalization phases, respectively). Hence the increase in the number of predictors did not result in a significant reduction in explained variance. In line with this result, the parameter estimates for the CS+ centered, Gaussian component were not significantly different from zero neither in baseline or generalization phases ($w_{Gaussian}$ = −0.009 ± 0.034 in baseline, $t(73)$ = −0.27, p=0.78; $w_{Gaussian}$ = 0.02 ± 0.02 in generalization, $t(73)$ = 0.74, p=0.46; *Figure 3C*). Also, pair-wise differences between parameter estimates did not reach significance ($t(73)$ = 0.72, p=0.47). We therefore conclude that further extensions of the Adversity Gradient model to include components for adversity-specific changes around the CS+, did not result in a better understanding of the adversity-induced changes in the similarity geometry of exploration strategies.

Due to our conditioning procedure, subjects explored CS+ and CS- faces more often than to the other faces. Therefore, the increase found in the specific component could be explained by repeated exposure of these faces. We tested this view with two different approaches. First, we evaluated whether the observed anisotropy ($w_{specific}$ - $w_{unspecific}$) in the similarity geometry was relevant for the aversive quality associated with the CS+ face. We hypothesized that stronger anisotropy could predict stronger aversive learning, as measured with the modulation depth of fear-tuning profiles coming from subjective ratings and skin-conductance responses. We found weak, but significant evidence for an association with an increased tuning strength in the ratings (r = 0.25, p=0.03), which was only marginally present for SCR (r = 0.23, p=0.07) (*Figure 3—figure supplement 1*). Second, if the increase in the specific component from baseline to test was merely driven by the exposure during the conditioning phase, the unspecific component would also increase when subjects were repeatedly exposed to the stimuli of the unspecific component over the three runs of the generalization phase. However, the weights of the unspecific component did not increase when analyzed for separate runs, but if anything showed a trend to decrease with further exposure (*Figure 3—figure supplement 2*). Both findings suggest that separation of exploration patterns along the adversity gradient was related to aversive learning and not to exposure differences.

## Spatial changes in exploration strategy

While the FPSA detected increased similarity in eye-movement patterns dominantly along the specific component, it aggregates information across all spatial units and does therefore not give information where and how subjects sample information differently after aversive learning. We thus sought to complement our FPSA analysis in a model-free way and to further elucidate how spatial exploration strategies changed. To this end, we aimed to test whether viewing behavior was predictive of whether subjects observed CS+ or CS- stimuli (or a pair of faces along the orthogonal dimension). We trained support vector machines (SVM, *Figure 4A*) to decode CS+ from CS- trials and −90° and +90° trials in each subject and tested whether classification accuracy differed between faces along the adversity gradient and orthogonal to the adversity gradient. This analysis can be linked to the found anisotropy of eye-movements, as the anisotropy suggests the classification accuracy along the specific dimension to exceed accuracy along the unspecific dimension due to stronger dissimilarities.

SVMs were able to decode CS+ vs CS- trials with an accuracy of 56% in the generalization phase compared to 50% in the baseline phase (*Figure 4B*). The unspecific direction (−90° vs 90°) could be decoded with an accuracy of 54% during the generalization phase and 52% during the baseline phase. A repeated measures ANOVA showed a significant interaction of generalization phase and component ($F(72)$ = 5.4, p=0.02) and a main effect of phase ($F(73)$ = 16.8, p=0.0001). These effects are compatible with FPSA results and reinforce the notion that changes were subtle. We also tested whether training SVMs across subjects instead of within subjects would reproduce this effect. Here the interaction between generalization phase and component was not significant ($F(73)$=0.43, p=0.51), suggesting that changes in viewing behavior are idiosyncratic and not necessarily comparable across subjects. Compatible with this, a traditional ROI based count analysis also showed no results (see below).

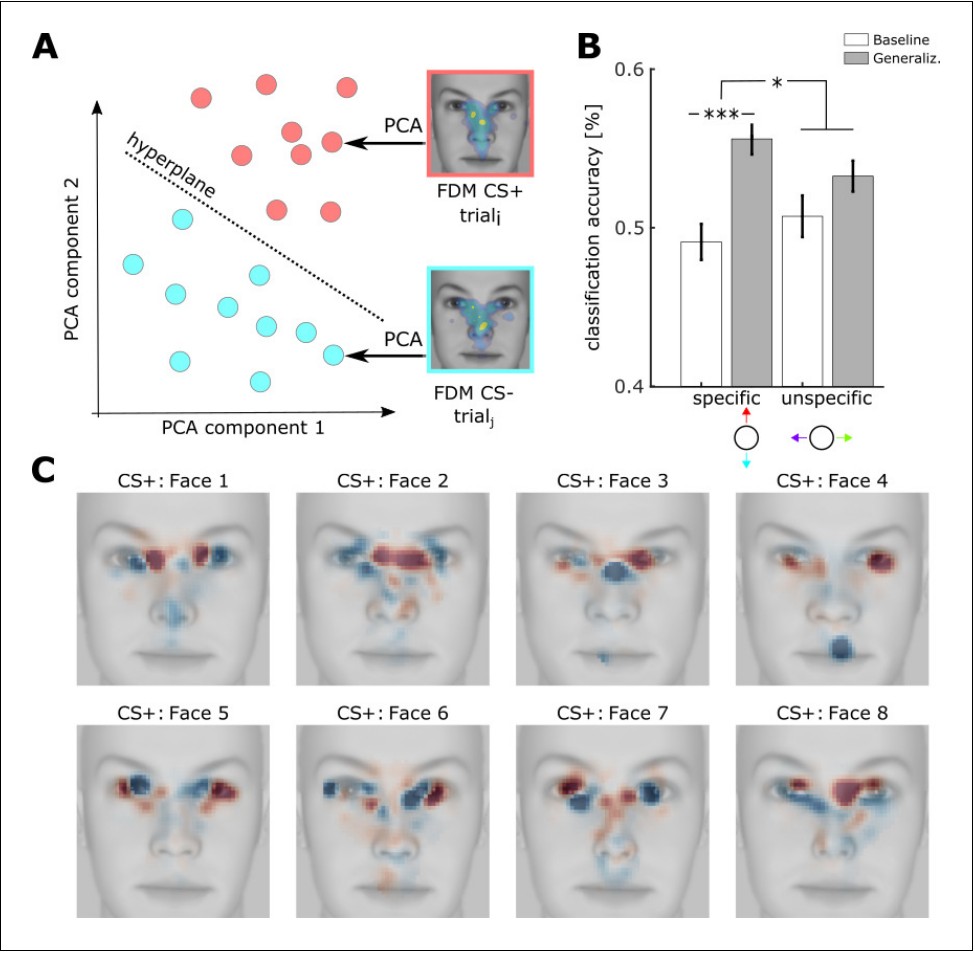

**Figure 4.** Classification of FDMs using support vector machines. (**A**) Classification of single trial FDMs using support vector machines. Before training, single FDMs of CS+ and CS- trials were reduced in their dimensionality (pixels) by a principal component analysis (PCA), resulting in a new representation in this feature space (red and cyan dots). A SVM was then trained to decode CS+ from CS- trials by finding a multidimensional hyperplane (dotted lines) separating CS+ from CS- patterns. The same was repeated for stimuli from the unspecific dimension (not shown). (**B**) Accuracy of SVM classification along the specific and unspecific dimension, trained within- subject for baseline (white) and generalization phase (gray) (M ± SEM, \*\*\*: $t(73)$ = 4.8, $p<0.001$, \*: $t(73)$ = 2.1, $p<0.05$ paired *t-test*) (see *Source code 1*) (**C**) Activation patterns derived from hyperplanes of classification of CS+ vs CS- trials as shown in (**A**). Activation patterns were z-scored and averaged across subjects with the same underlying physical CS+ face$_i$ ($n_i$ = [10, 12, 8, 10, 8, 10, 8, 8]), then superimposed on the respective face stimulus (see *Source code 1*).
DOI: https://doi.org/10.7554/eLife.44111.010
The following figure supplement is available for figure 4:

**Figure supplement 1.** Physical differences in opposing faces.
DOI: https://doi.org/10.7554/eLife.44111.011

Next, we were interested to see which locations in a face were informative for the classification process. To this aim, we extracted hyperplanes from each subjects' classification, converted them to activation patterns (*Haufe et al., 2014*) and averaged activation patterns of subjects with the same CS+ face. *Figure 4* shows these activation patterns superimposed on different physical faces. Qualitatively speaking activation patterns were remarkably different across faces. Furthermore, comparing activation patterns on a CS+/CS- combination with those activation patterns that have the opposite CS-/CS+ combination (compare top and bottom row in *Figure 4C*), suggested that these pairs are to some extent symmetric. In any case, all activation maps showed subtle changes along the eye region that qualitatively varied with the combination of CS+ and CS- face. Yet, physical differences between stimuli were not only present in the eye region, but spanned the whole face (*Figure 4—*

*figure supplement 1*). Moreover, these physical differences were also present for decoding along the orthogonal dimension, for which classification results were lower. Therefore, sampling more differently within the eye region could not be explained simply by the physical information they held.

## Temporal and spatial unfolding of adversity-specific exploration

While SCRs and subjective ratings provide insights about the aggregate cognitive evaluations of a given stimulus, eye-movements have the potential to provide information on how these cognitive evaluations unfold over both spatial and temporal domains. Aiming to explore the changes induced from aversive learning further with regards to these dimensions, we repeated the FPSA and fitted the Adversity Gradient Model on data slices from different temporal or spatial windows. For each temporal or spatial slice, we computed an anisotropy index, corresponding to the difference between specific and unspecific model parameters ($w_{specific} - w_{unspecific}$), and statistically evaluated the difference between before and after learning. For this analysis, we focused on participants that were able to correctly identify the CS+ based on their shock expectancy ratings (CS+>CS–, n = 61).

For the temporal analysis, we used a moving window of 500 ms with steps of 50 ms and (*Figure 5A*). While the time-course of adversity-specific and unspecific components was not distinguishable before learning, they diverged rather early in the subsequent generalization phase. The difference in anisotropy reached significance first at the time window corresponding to the interval 400–900 ms after stimulus onset (*Figure 5A* top row, paired t-test, p=0.03). As humans explore visual scenes serially with fixational eye-movements, the order of fixations (1st fixation, 2nd fixation, and so on) is another natural metric to evaluate temporal progress (*Tatler et al., 2005*). The same analysis indicated that adversity-specific exploration started following the first fixation (note that the first fixation is the landing fixation on the face following stimulus onset; *Figure 5A*) and stayed constant during the stimulus presentation. Overall, the temporal FPSA indicated that humans started to forage for adversity-relevant information early, as soon as after the first landing fixation.

We ran spatial FPSA at localized portions of the FDMs in a similar manner to a searchlight analysis in brain imaging (*Kriegeskorte et al., 2007*). For a given spatial portion (defined by a square window of 30 pixels,~1 visual degree), we fitted the Adversity Gradient Model, and assigned specific and unspecific weights to the center position of the searchlight (*Figure 5B*). The map that is obtained by repeating this analysis at all spatial locations provides an indication of the facial locations that are explored either with a specific or unspecific exploration strategy. We found that both before and after learning, specific and unspecific components were strongly localized around the eye region (*Figure 5B*). We tested the difference in anisotropy between the baseline and generalization phase within the three commonly used regions of interest at different facial elements (eyes, nose and mouth; ROIs shown in *Figure 5C*) (*Hessels et al., 2016*; *Malcolm et al., 2008*; *Schurgin et al., 2014*; *Walker-Smith et al., 1977*). We found this interaction to be significant only in the eye region (*Figure 5C*; paired t-test uncorrected, p<0.024). This result could be explained by an increased number of fixations around the eye region following learning and potentially result in a stronger signal-to-noise ratio for parameter estimation (*Diedrichsen et al., 2011*). Indeed, the eye region in this study accounted for ~72% of all fixations on the face . However, the increased anisotropy during generalization occurred despite a decrease of ~5% in fixation density around the same region (*top row* in *Figure 5—figure supplement 1*). This precludes the trivial explanation in terms of increased saliency of the eye region as a potential explanation for differences in similarity components. Overall, searchlight FPSA indicated that exploration strategies were specifically tailored to forage for adversity-specific information around the eye region, which corroborated the findings obtained by the SVM analysis described above.

## Comparison of FPSA to common eye-movement features and ROI-based analyses

As our model-based FPSA as well as machine-learning approaches are rather complex analyses, we aimed to compare these methods to classical approaches used for the analysis of eye movements. In an explorative approach, we tested whether aversive learning induced changes in classical measures of viewing behavior, such as changes in the number of fixations, their duration, saccade length and the entropy of individual FDMs for all eight faces. To be able to compare these features across subjects, they were z-scored within phase and subject, then averaged across subjects to check for

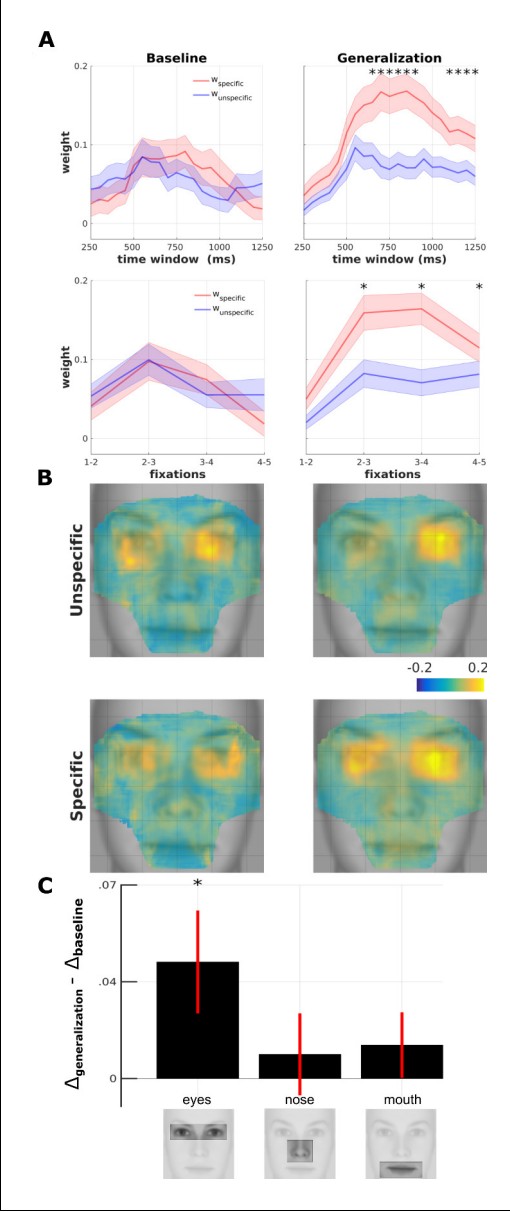

**Figure 5.** Spatio-temporal fixation-pattern similarity analysis. (**A**) Temporal development of adversity-specific and unspecific exploration strategies (n = 61). Parameters of the adversity categorization model (*red*: specific; *blue*: unspecific) are computed using a moving time window of 500 ms at intervals of 50 ms for baseline (*left panels*) and generalization phase (*right panel*). Numbers on the x-axis denote the center of the moving window. Second row depicts the same analysis with fixation points sorted by rank. Asterisks indicate time points with statistically significant interaction testing for difference in anisotropy ($w_{specific} - w_{unspecific}$) between test vs. baseline (*: p<0.05; Shaded area: SEM) (**B**) Four maps resulting from the searchlight-FPSA on FDMs from before and after conditioning (*left vs. right columns)* and for unspecific and specific (*top vs. bottom rows*) model parameters overlaid on an

*Figure 5 continued on next page*

commonalities in these individual profiles. Descriptive plots are shown in *Figure 5—figure supplement 2*. We found that the number of fixations, the average saccade distance and entropy showed significant differences between CS+ and CS-, all of which were higher for the CS+. The only measure exhibiting a significant generalization profile, that is bell-shaped tuning curve, after aversive learning was the saccade length (p<0.001, log-Likelihood Ratio Test of Gaussian against flat null-model). While simple outcome measures such as fixation and saccade characteristics changed over the course of the experiment, their univariate nature prevents us from correcting for changes not induced by aversive learning - that is developed along the unspecific component. To control for this and to connect the findings to the anisotropy found by FPSA, we set up a linear model testing whether the anisotropy in these measures (i.e. specific – unspecific) could predict the anisotropy of eye movements in the subjects. However, none of the features was able to robustly predict anisotropy (F(4,69) = 0.075, p=0.55, $R^2_{adjusted}$ = -0.01).

We also compared FPSA on eye movement patterns to common ROI-based analyses on fixation counts. We computed changes in fixation counts in the three common regions of face stimuli, that is eyes, nose and mouth (same as depicted *Figure 5C*). If conditioning before the generalization phase lead to an increased saliency of facial features that are diagnostic of the CS+ face, one would expect a non-flat fear tuning in the number of fixations towards these facial features, which would receive more fixations with increasing similarity to the CS+ face. In line with previous reports (*Walker-Smith et al., 1977*), eyes together with the nose region were the most salient locations across the baseline and generalization phases, and attracted ~91% of all fixation density, whereas the mouth region had only a marginal contribution with ~3.5%. Investigating changes from baseline to generalization phase, we found that aversive learning increased the number of fixations directed at the nose (+3.6%) and mouth (+0.6%) regions at the expense of the eye region (–5.1%). Most importantly, model comparison on fixation density favored the flat null model for all regions even at low statistical thresholds in all facial elements across both baseline and generalization phases (p>0.05, log-likelihood test; *Figure 5—figure supplement 1*). While a weak tuning was apparent in the mouth region during generalization, this did not reach significance (p>0.1). Therefore, our observations at the group-level were limited

*Figure 5 continued*

average face. The map is masked to contain 90% of all fixation density. (C) Difference of anisoptropy between before and after aversive learning (*generalization – baseline*) for three different ROIs (*: p<0.01, paired t-test).

DOI: https://doi.org/10.7554/eLife.44111.012

The following figure supplements are available for figure 5:

**Figure supplement 1.** Generalization profiles on ROI-based fixation counts.

DOI: https://doi.org/10.7554/eLife.44111.013

**Figure supplement 2.** Generalization profiles of common fixation features.

DOI: https://doi.org/10.7554/eLife.44111.014

to unspecific changes in fixation density across phases, that were independent of the adversity gradient introduced through conditioning.

## Discussion

The present work tested how aversive learning changed the exploration of faces following aversive learning. We used a stimulus continuum that was defined by perceptual similarity, as well as adversity-related information as two independent perceptual factors. As an extension of similarity-based multivariate pattern analyses used to investigate representational content of neuronal activity in fMRI (*Kriegeskorte, 2008*) and MEG/EEG (*Cichy et al., 2014*; *Kietzmann et al., 2017*), we introduced FPSA to characterize learning-induced changes in eye-movement behavior during free viewing of faces. As expected, before aversive learning exploration patterns mirrored the inherent circularity of stimulus similarity. This is compatible with the view that the exploration of neutral faces in our experimental settings is, at least to some extent, guided by physical characteristics, in contrast to a completely holistic viewing strategy (*Peterson and Eckstein, 2012*). Aversive learning led to a specific increase in dissimilarity along the direction of the induced adversity gradient. This adversity-specific exploration strategy appeared early, as soon as the landing fixation, and lasted continuously for the duration of stimulus presentation.

It is an ongoing debate, what kind of information drives generalization after aversive learning. According to one prominent view, the perceptual model, the degree to which a novel stimulus is considered as harmful is directly related to the degree of overlap between shared sensory features with a previously learnt harmful stimulus. An alternative view proposes that fear generalization is an active cognitive act (*Shepard, 1987*), related specifically to the prediction of potential threat in uncertain situations (*Onat and Büchel, 2015*). According to the threat-prediction model, perceptual factors can contribute to fear generalization, but only to the extent they are predictive of harmful events (*Lashley and Wade, 1946*; *Shepard, 1987*; *Tenenbaum and Griffiths, 2001*; *Baddeley et al., 2007*; *Soto et al., 2014*). Yet, in the majority of previous studies, perceptual similarity between the generalization and learning samples has been explicitly used as a cue for signaling the threat. As a result, threat-prediction has commonly been confounded by perceptual similarity, making it impossible to dissociate their independent contributions. The finding that fixation strategies change incrementally along the dimension that is predictive of threat, in our case captured by the anisotropy of specific vs unspecific components, offers support for the threat-prediction model.

Still, there are at least two different scenarios that are compatible with a selective separation of exploration patterns for the CS+ and CS– faces (*Figure 6*). This regards the way how learning potentiates sensory features as being diagnostic for the prediction of harmful outcomes, thereby making them target locations for attentional allocation. In the first scenario, aversive learning potentiates combinations of visual features that are specific to the harm-predicting item, namely the CS+ face identity (*Figure 6A*, *square box*). In an alternative view, aversive learning consists of recovering the vector of features that defines the adversity gradient (*Figure 6B*, *black arrow*), leading to an increased saliency for discriminative features that separate best the harm- and safety-predicting prototypes. This feature vector could overlap with categorical knowledge that is either naturally present (such as gender, ethnicity or emotional expression), or learned with experience (*Dunsmoor and Murphy, 2015*; *Kietzmann and König, 2010*; *Qu et al., 2016*). While both scenarios lead to an increased separation between the CS+ and CS– poles, as observed in the present study, they have divergent predictions when tested with stimuli organized in three concentric circles (*Figure 6B*). Potentiation of identity-specific representations would lead faces that are similar to the CS+ on outer and inner circles to be explored similarly, resulting in a shrinkage of the similarity geometry around the CS+ face and thereby leading to a shift in the center of ellipses (*Figure 6B*, *right panel*). On the

other hand, if learning consists of forming a vector representation of the adversity gradient, the separation would add to differences that are already present, resulting in three concentric ellipses centered on the same point. Future investigations can distinguish these two hypotheses by testing the presence of a shift in the center of similarity geometry towards the CS+ face or not.

In either case, selective changes in the similarity of exploration patterns are compatible with theoretical work, that has shown that generalization can be viewed as the formation of an internal model of the environment in order to predict the occurrence of behaviorally relevant events. Shepard's generalization theory conceives generalization as the formation of a binary categorical zone along a smooth perceptual continuum for the prediction of harmful or safe events (*Shepard, 1987*). In this view, empirical generalization profiles can be understood as resulting from different stimuli to be registered into either the one or the other category in a probabilistic manner. In the same line, Bayesian frameworks refer explicitly to probability distributions to represent internal beliefs for the formation of generalization strategies (*Tenenbaum and Griffiths, 2001*; *Tenenbaum et al., 2006*). These theoretical works have already pointed to the possibility that humans can adaptively tailor generalization strategies by appropriately structuring their internal beliefs even in situations which are not defined by perceptual factors. Our results provide supporting evidence that these mechanisms have their validity even in settings that are defined solely by perceptual factors.

Introduction of the FPSA methodology was not a choice, but rather a necessity to test the outlined competing hypotheses. First, eye-movement patterns during free-viewing of faces are characterized by strong inter-individual variability (*Coutrot et al., 2016*; *Kanan et al., 2015*; *Mehoudar et al., 2014*; *Walker-Smith et al., 1977*). This was also true in the present study, where we could train linear classifiers to predict a volunteer's identity with an average accuracy of ~80% based on fixation maps (results not shown). This is possibly one major reason why we

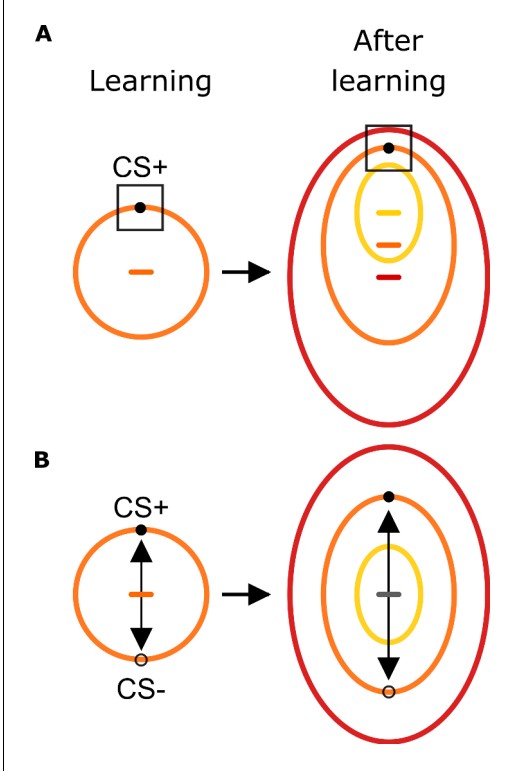

**Figure 6.** Predictions on the influence of aversive learning on sensory-motor foraging strategies. Before learning, exploration strategies follow the circularity of the stimulus space (*orange circle*) in line with their presumed physical characteristics. (**A**) In one scenario, conditioning leads humans to learn the specific feature values that predict a harmful outcome (*black square*). When tested subsequently with stimuli organized as three concentric stimulus gradients, this scenario predicts faces that are similar to the CS+ face to be explored similarly. This results in a global shift in the center of gravity towards the CS+ face as indicated by *yellow, orange* and *red horizontal dashes* indicating the center of corresponding ellipses. (**B**) Humans learn the feature vector (*black arrow*) that best separates harmful and safety predicting stimuli. When tested with the concentric circular stimulus set, this scenario predicts three concentric ellipses sharing the same center of gravity (*gray horizontal line*).

DOI: https://doi.org/10.7554/eLife.44111.015

did not observe adversity-tuned generalization profiles at different facial ROIs (*Figure 5—figure supplement 1*). While this inter-individual variability in exploration patterns can dilute the sensitivity of ROI-based approaches, FPSA exploits consistent changes in the similarity relationships of exploration patterns that were defined within-subject. Second, and equally importantly, using multivariate responses allowed to assess changes in similarity between all pairs of stimuli. It is not clear how the same could be achieved with subjective ratings or SCR measurements that are collapsed over time.

For humans, faces are a rich source of information for judging potential outcomes in social interactions. Face stimuli were thus a natural choice to investigate how active exploration strategies change with learning. Together with the FPSA method, a two dimensional circular continuum of

faces allowed us to test different hypotheses that would have not been easily possible using a one-dimensional perceptual gradient. However, the circular continuum was based on two arbitrary dimensions (identity and gender). Any prior experiences that participants had in real-life situations with these dimensions might potentially bias our results. For example, males could generally be perceived more dangerous than females (*Navarrete et al., 2009*). Therefore, any real-life categories present in the stimulus set might induce prior expectations about the potential harmfulness that have to be relearned during the aversive conditioning process. Yet, as the face chosen as CS+ was carefully counterbalanced across participants, we consider the individual impact of arbitrarily chosen dimensions to cancel out across subjects.

On the group-level, we found the adversity gradient model to be most compatible with observed changes in exploration patterns. While the majority of participants was in line with this view, on the individual level, many participants actually did not exhibit the effect of interest. We tested a possible heterogeneity of our participant sample, which was not supported by a Gaussian mixture model analysis. Future investigations should aim to understand whether this type of subtle effects results from motivational aspects during this type of learning tasks, or a genuine source of noise in the adversity learning and its interaction with exploration behavior. In any case, the rather small effects call for replication out of sample, as well as further investigations testing the predictions derived from our data. The above mentioned design of multiple circles as stimulus material (*Figure 6*) would allow to replicate, verify or falsify the predictions made in this study, as three circles would allow more specific and stronger predictions that could then be tested on an independent sample.

Could our results be explained by an unbalanced exposure to the CS+ and CS– faces that were presented during the conditioning phase? As participants have seen these faces more often, one can argue that this could potentially bias eye movement patterns. While we cannot completely exclude a contribution of exposure, we have shown that anisotropy correlates with indicators of aversive learning as measured by subjective ratings and, to lesser extent with autonomous activity. Furthermore, the unspecific component did not increase with exposure over multiple runs, which would be expected if it was solely driven by exposure. Therefore, we believe that the major drive that leads to the separation of patterns along the specific axis are due to the affective nature of learning, rather than occurring merely as a result of exposure.

Eye-movements patterns can provide important insights about what the nervous system tries to achieve as they summarize the final outcome of complex interactions at the neuronal level (*König et al., 2016*). Our results demonstrate that changes induced by aversive generalization extend beyond autonomous systems or explicit subjective evaluations, but can also affect an entire sensory-motor loop at the systems level (*Dowd et al., 2016*). Furthermore, the methodology applied here can easily be extended to neuronal recordings, where gradients of activity during generalization have been successfully used to characterize selectivity of aversive representations. Therefore, it will be highly informative to test different hypotheses we outlined here using neuronal recordings with representational similarity analysis during the emergence of aversive representations.

## Materials and methods

### Participants

Participants were 74 naïve healthy males and females (n = 37 each) with normal (or corrected-to-normal) vision (age = 27 ± 4, M ± SD) and without history of psychiatric or neurological diseases, any medical condition or use of medication that would alter pain perception. Participants had not participated in any other study using facial stimuli in combination with aversive learning before. They were paid 12 Euros per hour for their participation in the experiment and provided written informed consent. All experimental procedures were approved by the Ethics committee of the General Medical Council Hamburg.

### Data sharing

The dataset used in this manuscript has been published as a dataset publication (*Wilming et al., 2017*). We publicly provide the stimuli as well as the Matlab (MathWorks, Natick MA) code necessary for the reproduction of all the results including figures presented in this manuscript (*Onat and*

*Kampermann, 2017*). The code can be used to download the data as well. Code used for linear support vector machines (see below) is accompanying this publication as *Source code 1*.

## Stimulus preparation and calibration of generalization gradient

Using a two-step procedure, we created a final set of 8 calibrated faces (*Figure 1A*, see also *Figure 1—figure supplement 1*) that were perceptually organized along a circular similarity continuum based on a model of the primary visual (V1) cortex. Using the FaceGen software (FaceGen Modeller 2.0, Singular Inversion, Ontario Canada) we created two gender-neutral facial identities and mixed these identities (0%/100% to 100%/0%) while simultaneously changing the gender parameters in two directions (more male or female). In the first step, we created a total of 160 faces by appropriately mixing the gender and identity parameters to form five concentric circles (see *Figure 1—figure supplement 1*) based on FaceGen defined parameter values for gender and identity. Using a simple model of the primary visual cortex known to closely mirror human perceptual similarity judgments (*Yue et al., 2012*), we computed V1 representations for each face after converting them to grayscale. The spatial frequency sensitivity of the V1 model was adjusted to match human contrast sensitivity function with bandpass characteristics between 1 and 12 cycles/degree, peaking at six cycles/degrees (*Blakemore and Campbell, 1969*). The V1 model consists of pair of Gabor filters in quadrature at five different spatial scales and eight orientations. The activities of these 40 channels were averaged in order to obtain one single V1 representation per face. We characterized the similarity relationship between the V1 representations of 160 faces using multidimensional scaling analysis with two dimensions (*Figure 1—figure supplement 2*). As expected, while two dimensions explained a large variance, the improvement with the addition of a third dimension was only minor, providing thus evidence that the physical properties of the faces were indeed organized along two-dimensions (stress values for 1D, 2D and 3D resulting from the MDS analysis were 0.42, 0.04, 0.03, respectively). The transformation between the coordinates of the FaceGen software values (gender and identity mixing values) and coordinates returned by the MDS analysis allowed us to gather FaceGen coordinates that would correspond to a perfect circle in the V1 model. In the second step, we thus generated eight faces that corresponded to a perfect circle. This procedure ensured that faces used in this study were organized perfectly along a circular similarity continuum according to a simple model of primary visual cortex with well-defined bandpass characteristics known to mirror human similarity judgments. Furthermore, it ensured that dimensions of gender and identity introduced independent variance on the faces.

To present these stimuli we resized them to 1000 × 1000 pixels (originals: 400 × 400) using bilinear interpolation, and slightly smoothed with a Gaussian kernel of 5 pixels with full-width at half maximum of 1.4 pixels to remove any possible pixel artifacts that could potentially lead participants to identify faces. Faces were then normalized to have equal luminance and root-mean-square contrast. The gray background was set to the same luminance level ensuring equal brightness throughout of the experiment. Faces were presented on a 20' monitor (1600 × 1200 pixels, 60 Hz) using Matlab R2013a (Mathworks, Natick MA) with psychophysics toolbox (*Brainard, 1997*; *Pelli, 1997*). The distance of the participants' eyes to the stimulus presentation screen was 50 cm. The center of the screen was at the same level as the participants' eyes. Faces spanned horizontally ~17° and vertically ~30°, aiming to mimic a typical face-to-face social situation. Stimuli are available in *Onat and Kampermann (2017)*.

## Experimental paradigm

The fear conditioning paradigm (similar to *Onat and Büchel, 2015*) consisted of baseline, conditioning and test (or generalization) phases (*Figure 2A*). Participants were instructed that the delivery of UCSs during baseline would not be associated with faces, however in the following conditioning and generalization phases they were instructed that shocks would be delivered after particular faces have been presented. In all three phases, subjects were instructed to press a button when an oddball stimulus appeared on the screen.

Four equivalent runs with exactly same number of trials were used during baseline (one run) and generalization phases (three runs) consisting of 120 trials per run (~10 min). Every run started with an eye-tracker calibration. Between runs participants took a break and continued with the next run in a self-paced manner. We avoided having more than one run in the baseline period in order not to

induce fatigue in participants. At each run during the baseline and generalization phases, eight faces were repeated 11 times, UCS trials occurred five times and one oddball was presented. This consisted of a blurred unrecognizable face, which volunteers were instructed to press a key. We presented 26 null trials with no face presentation but otherwise the same trial structure (see below sequence optimization). In order to keep arousal levels comparable to the generalization phase, UCSs were also delivered during baseline, however they were fully predictable by a shock symbol therefore avoiding any face to UCS associations.

During the conditioning phase, participants saw only the CS+ and the CS– faces (and null trials). These consisted of 2 maximally dissimilar faces separated by 180° on the circular similarity continuum and randomly assigned for every participant in a balanced manner. The conditioning was 124 trials long (~10 min) and CS+ and CS– faces were repeated 25 times. CS+ faces were additionally presented 11 times with the UCSs, resulting in a reinforcement rate of ~30%. The same reinforcement ratio was used during the subsequent generalization phase in order to avoid extinction of the learnt associations.

Stimulus presentation sequence was optimized for the deconvolution of the skin-conductance responses. This regards the choice of conditions and the duration of interstimulus intervals. Faces were presented using a rapid-event design with a stimulus onset asynchrony of 6 s and stimulus duration of 1.5 s. The presentation sequence was optimized using a modified m-sequence with 11 different conditions (*Buracas and Boynton, 2002*; *Liu and Frank, 2004*) (eight faces, UCS, oddball, null). An m-sequence is preferred as it balances all transitions from condition $n$ to $m$ (thus making the sequence as unpredictable as possible for the participant) while providing an optimal design efficiency (thus making deconvolution of autonomic skin conductance responses more reliable). However, all conditions in an m-sequences appear equally number of times. Therefore, in order to achieve the required reinforcement ratio (~30%), we randomly pruned UCS trials and transformed them to null trials. Similarly, oddball trials were pruned to have an overall rate of ~1%. This resulted in a total of 26 null trials. While this deteriorated the efficiency of the m-sequence, it was still a good compromise as the resulting sequence was much more efficient than a random sequence. Resulting from the intermittent null trials, SAOs were 6 or 12 s approximately exponentially distributed.

Face onsets were preceded by a fixation-cross, which appeared randomly outside of the face either on the left or right side along an virtual circle (r = 19.6°, + /- 15° above and below the horizontal center of the image). The side of fixation-cross was balanced across conditions to avoid confounds that might occur (*Arizpe et al., 2015*). Therefore, the first fixation consisted of a landing fixation on the face.

## Calibration and delivery of electric stimulation

Mild electric shocks were delivered by a direct current stimulator (Digitimer Constant Current Stimulator, Hertfordshire UK), applied by a concentric electrode (WASP type, Speciality Developments, Kent UK) that was firmly connected to the back of the right hand and fixated by a rubber glove to ensure constant contact with the skin. Shocks were trains of 5 ms pulses at 66 Hz, with a total duration of 100 ms. During the experiment, they were delivered right before the offset of the face stimulus. The intensity of the electric shock applied during the experiment was calibrated for each participant before the start of the experiment. Participants underwent a QUEST procedure (*Watson and Pelli, 1983*) presenting UCSs with varying amplitudes selected by an adaptive algorithm and were required to report whether a given trial was 'painful' or 'not painful' in a binary fashion using a sliding bar. The QUEST procedure was repeated twice to account for sensitization/habituation effects, thus obtaining a reliable estimate. Each session consisted of 12 stimuli, starting at an amplitude of 1mA. The subjective pain threshold was the intensity that participants would rate as 'painful' with a probability of 50%. The amplitude used during the experiment was two times this threshold value. Before starting the actual experiment, participants were asked to confirm whether the resulting intensity was bearable. If not then the amplitude was incrementally reduced and the final amplitude was used for the rest of the experiment.

## Eye tracking and fixation density maps

Eye tracking was done using an Eyelink 1000 Desktop Mount system (SR Research, Ontario Canada) recording the right eye at 1000 Hz. Participants placed their head on a headrest supported under

the chin and forehead to keep a stable position. Participants underwent a 13 point calibration/valida-tion procedure at the beginning of each run (1 Baseline run, 1 Conditioning run and 3 runs of Gener-alization). The average mean-calibration error across all runs was Mean = 0.36°, Median = 0.34°, SD = 0.11. 91% of all runs had a calibration better than or equal to. 5°.

Fixation events were identified using commonly used parameter definitions (*Wilming et al., 2017*) (Eyelink cognitive configuration: saccade velocity threshold = 30°/second, saccade accelera-tion threshold = 8000° per second$^2$, motion threshold = 0.1°). Fixation density maps (FDMs) were computed by spatially smoothing (Gaussian kernel of 1° of full width at half maximum) a 2D histo-gram of fixation locations, and were transformed to probability densities by normalizing to unit sum. FDMs included the center 500 × 500 pixels, including all facial elements where fixations were mostly concentrated (~95% of all fixations).

## Shock expectancy ratings and autonomic recordings

After baseline, conditioning and generalization phases, participants rated different faces for subjec-tive shock expectancy by answering the following question, '*How likely is it to receive a shock for this face?*''. Faces were presented in a random order and rated twice. Subjects answered using a 10 steps scale ranging from '*very unlikely*' to '*very likely*' and confirmed by a button press in a self-paced manner.

Electrodermal activity evoked by individual faces was recorded throughout the three phases. Reusable Ag/AgCl electrodes filled with isotonic gel were connected to the palm of the subject's left hand using adhesive collars, placed in thenar/hypothenar configuration. Skin-conductance responses were continuously recorded using a Biopac MP100 AD converter and amplifier system at a sampling rate of 500 Hz. Using the Ledalab toolbox (*Benedek and Kaernbach, 2010b*; *Benedek and Kaernbach, 2010a*), we decomposed the raw data to phasic and tonic response com-ponents after downsampling it to 100 Hz. Ledalab applies a positively constrained deconvolution technique in order to obtain phasic responses for each single trial. We averaged single-trial phasic responses separately for each condition and experimental phase to obtained 21 average values (9 (8 faces + one null condition) from baseline and generalization and 3 (2 faces + one null condition) from the conditioning phase). CS+ trials with UCS were excluded from this analysis. These values were first log-transformed ($\log_{10}(1+SCR)$) and subsequently z-scored for every subject separately (across all conditions and phases), then averaged across subjects. Therefore, negative values indicate phasic responses that are smaller than the average responses recorded throughout the experiment. Due to technical problems, SCR data could only be analyzed for n = 63 out of the 74 participants.

## Nonlinear modeling and model comparison

We fitted a Gaussian function to generalization profiles obtained from subjective ratings, skin-con-ductance responses and fixation counts at different ROIs by minimizing the following likelihood term in (1) following an initial grid-search for parameters

$$L(D(x)|\theta, \sigma) = \Sigma - \log[N(D(x) - G(x|\theta)|0, \sigma)] \qquad (1)$$

where *x* represents signed angular distances from a given volunteer's CS+ face; G(*x*|θ) is a Gaussian function that was used to model the adversity tuning. It is defined by the parameter vector θ, which codes for the amplitude (difference between peak and base) and width of the resulting generaliza-tion profile; D(*x*) represents the observed generalization profile for different angular distances; and N(*x*| 0, σ) is the normal probability density function with mean zero and standard deviation of σ. The fitting procedure consisted of finding parameters' values that minimized the sum of negative log-transformed probability values. Using log-likelihood ratio test we tested whether this model per-formed better than a null model consisting of a horizontal line, effectively testing the significance of the additional variance explained by the model. G(x) was in the form

$$G(x) = \alpha \exp(-(x./\sigma_G)^2/2) \qquad (2)$$

$\alpha$ represents the depth of adversity tuning which corresponds to the difference between peak and baseline responses, $\sigma_G$ controls the width of the tuning.

As Gaussian shaped generalization profiles on the group level can equally result from binary pro-files on subject level, we compared model fits of individual Gaussian tunings to binary profiles. The

binary profile was defined by an amplitude parameter $\alpha$, width and offset (base value) parameter and can be found at Tuning.binary_freeY at *Onat and Kampermann (2017)*.

When comparing model-fits of these two profiles in individual subjects, we compare the likelihoods $L(D(x) \mid \theta, \sigma)$ of Gaussians to binary fits in each subject using the Akaike Information Criterion (AIC) to penalize for the more complex binary function ($df_{bin}$ = 3 vs. $df_{Gauss}$ = 2).

## Fixation-pattern similarity analysis

FPSA was conducted on single participants. Condition specific FDMs (eight faces per baseline and generalization phases) were computed by collecting all fixations across trials on a single map which was then normalized to unit sum. We corrected FDMs by removing the common mean pattern (done separately for baseline and generalization phases). We used 1 - Pearson correlation as the similarity metric. This resulted in a 16 × 16 similarity matrix per subject. Statistical tests for element-wise comparison of the similarity values were conducted after Fisher transformation of correlation values. The multidimensional scaling was conducted on the baseline and generalization phases jointly using the 16 × 16 similarity matrix as input (*mdscale* in MATLAB). Importantly, as the similarity metric is extremely sensitive to the signal to noise ratio (*Diedrichsen et al., 2011*) present in the FDMs, we took precautions that the number of trials between generalization and baseline phases were exactly the same in order to avoid differences that would have been caused by different signal to noise ratios. To account for unequal number of trials during the baseline (11 repetitions) and generalization (3 runs x 11 = 33 repetitions) phases, we computed a similarity matrix for each run separately in the generalization phase. These were later averaged across runs for a given participant. This ensured that FDMs of the baseline and generalization phases had comparable signal-to-noise ratios, therefore not favoring the generalization phase for having more trials.

We generated three different models based on a quadrature decomposition of a circular similarity matrix. A circular similarity matrix of 8 × 8 can be obtained using the term $\mathbf{M} \otimes \mathbf{M}$, where $\mathbf{M}$ is a 8 × 2 matrix in form of [cos(x) sin(x)], and the operator $\otimes$ denotes the outer product. $x$ represents angular distances from the CS+ face, is equal to 0 for CS+ and $\pi$ for CS–. Therefore, while cos(x) is symmetric around the CS+ face, sin(x) is shifted by 90°. For the Perceptual Baseline and Perceptual Expansion models (*Figure 1B and C*) we used $\mathbf{M} \otimes \mathbf{M}$ as a predictor together with a constant intercept. For the Adversity Gradient model depicted in *Figure 1D*, we used cos(x) $\otimes$cos(x) and sin(x)$\otimes$sin(x) to independently model ellipsoid expansion along the specific and unspecific directions, respectively. Together with the intercept this model comprised three predictors. Finally, the CS+ Attraction model (*Figure 1E*) was created using the predictors of the tuned exploration model in conjunction with a two-dimensional Gaussian centered on the CS+ face (in total four predictors). We tested different widths for the Gaussian and took the one that resulted in the best fit. This was equal to 65° of FWHM and similar to the values we observed for univariate explicit ratings and SCR responses.

All linear modeling was conducted using non-redundant, vectorized forms of the symmetric dissimilarity matrices. For an 8 × 8 dissimilarity matrix this resulted in a vector of 28 entries. Different models were fitted as mixed-effects, where intercept and slope contributed both as fixed- and random-effects (*fitlme* in Matlab). We selected mixed-effect models as these performed better than models defined uniquely with fixed-effects on intercept and slope. To do model selection, we used Bayesian information criterion (BIC) as it compensates for an increase in the number of predictors between different models. Additionally, different models were also fitted to single participants (*fitlm* in Matlab) and the parameter estimates were separately tested for significance using *t-test*.

For the analysis of temporal and spatial unfolding of adversity specific exploration patterns, the same analysis was run, but restricted to include only the given time windows/fixations. In this analysis, we only included participants who had a significant fear-tuning of explicit shock ratings from the generalization phase (n = 61). For the time-windowed approach, periods of 500 ms were used, repeated shifted by 50 ms, thereby obtaining a rolling-window analysis, while fixation-wise analyses were based on FDMs that included only on the given (1st, 2nd, and so on) fixation.

## Classification with linear support vector machine

We used single trial FDMs for validation of linear support vector machines (*Chang and Lin, 2011*), that were trained to classify exploration patterns obtained during viewing of CS+ and CS– faces or orthogonal faces. Analyses using SVM were executed using scikit learn and python

(*Pedregosa et al., 2011*). The code has been included as *Source code 1* accompanying this publication. To reduce the dimensionality, FDMs were first downscaled 10 times resulting in a vector of 2500 pixels and by discarding pixels whose standard deviation across all conditions and subjects was much larger than its mean (mean/std > 0.075). We further reduced dimensionality by projecting FDMs onto their principal components using N principal components that explained 50% of the total variance in each run. SVMs were trained with l2 regularization and for each individual subject to account for idiosyncrasy in scanning patterns. We used 3-fold cross validation and ensured that all parameters were exclusively estimated from training data. In total we evaluated four SVMs per subject, two with data from the baseline phase and two with data from the generalization phase. For visualization purposes, SVMs were also trained on 100% of the trials. We converted resulting hyperplanes to activation patterns (*Haufe et al., 2014*) and averaged across subjects that shared the same combination of CS+ and CS- faces.

## Acknowledgements

The authors wish to thank Tim Kietzmann for his input on an early version of this manuscript, Cliodhna Quigley for proof-reading, Helen Blank for comments, Lukas Neugebauer for helpful input on the comparison of different tuning profiles, Patricia Billaudelle and Katrin Harland for their assistance with data collection. Selim Onat is supported by the DFG SFB TRR 58.

## Additional information

### Competing interests

Christian Büchel: Reviewing editor, *eLife*. The other authors declare that no competing interests exist.

### Funding

| Funder | Grant reference number | Author |
|---|---|---|
| Deutsche Forschungsgemeinschaft | SFB TRR 58 | Lea Kampermann<br>Christian Büchel<br>Selim Onat |
| Deutsche Forschungsgemeinschaft | SFB 936/A7 | Niklas Wilming |
| H2020 Marie Skłodowska-Curie Actions | 753441 | Arjen Alink |

The funders had no role in study design, data collection and interpretation, or the decision to submit the work for publication.

### Author contributions

Lea Kampermann, Software, Formal analysis, Validation, Investigation, Visualization, Methodology, Writing—original draft, Writing—review and editing; Niklas Wilming, Formal analysis, Validation, Methodology, Writing—review and editing; Arjen Alink, Validation, Writing—review and editing; Christian Büchel, Conceptualization, Resources, Funding acquisition, Validation, Writing—review and editing; Selim Onat, Conceptualization, Data curation, Software, Formal analysis, Supervision, Validation, Visualization, Methodology, Writing—original draft, Project administration, Writing—review and editing

### Author ORCIDs

Lea Kampermann  https://orcid.org/0000-0001-9016-6212
Niklas Wilming  http://orcid.org/0000-0003-0663-9828
Arjen Alink  http://orcid.org/0000-0001-6468-5449
Christian Büchel  http://orcid.org/0000-0003-1965-906X
Selim Onat  https://orcid.org/0000-0002-4782-5603

## Ethics

Human subjects: All experimental procedures were approved by the Ethics committee of the General Medical Council Hamburg (PV 4164). Participants had not participated in any other study using facial stimuli in combination with aversive learning before. They were paid 12 Euros per hour for their participation in the experiment and provided written informed consent.

## Decision letter and Author response

Decision letter https://doi.org/10.7554/eLife.44111.024
Author response https://doi.org/10.7554/eLife.44111.025

# Additional files

## Supplementary files

• Source code 1. Code implementing classification of single trials with linear support vector machines (see *Figure 4*).
DOI: https://doi.org/10.7554/eLife.44111.016

• Supplementary file 1. Supplementary tables reporting mixed-effects modeling of the three models shown in *Figure 1B–E*. (**A**) Mixed-effects modeling of the similarity matrices during the baseline phase with the Perceptual model shown in *Figure 1B*. (**B**) Mixed-effects modeling of the similarity matrices during the generalization phase with the Perceptual model shown in *Figure 1B*. (**C**) Mixed-effects modeling of the similarity matrices during the baseline phase with the Adversity Gradient model shown in *Figure 1D*. (**D**) Mixed-effects modeling of the similarity matrices during the generalization phase with the Adversity Gradient model shown in *Figure 1D*. (**E**) Mixed-effects modeling of the similarity matrices during the baseline phase with the CS+ Attraction model shown in *Figure 1E*. (**F**) Mixed-effects modeling of the similarity matrices during the generalization phase with the CS+ Attraction model shown in *Figure 1E*.
DOI: https://doi.org/10.7554/eLife.44111.017

• Transparent reporting form
DOI: https://doi.org/10.7554/eLife.44111.018

## Data availability

All source data and analysis- and figure-generating scripts are available for download from the project home page at the Open Science Framework (at https://osf.io/zud6h/). Source code for the analyses shown in Figure 4 has been added to the manuscript as Source code 1.

The following datasets were generated:

| Author(s) | Year | Dataset title | Dataset URL | Database and Identifier |
|---|---|---|---|---|
| Wilming N, Onat S, Ossandón J, Acik A, Kietzmann TC, Kaspar K, Gameiro RR, Vormberg A, König P | 2017 | Data from: An extensive dataset of eye movements during viewing of complex images | https://doi.org/10.5061/dryad.9pf75 | Dryad Digital Repository, 10.5061/dryad.9pf75 |
| Onat S, Kampermann L | 2017 | Similarity Analysis of Fixation Patterns during Aversive Generalization | https://doi.org/10.17605/OSF.IO/ZUD6H | Open Science Framework, 10.17605/OSF.IO/ZUD6H |

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
