## [Decision Letter]

Thank you for submitting your article "Threat prediction as the relevant factor for understanding fear generalization" for consideration by *eLife*. Your article has been reviewed by three peer reviewers, one of whom is a member of our Board of Reviewing Editors, and the evaluation has been overseen by Richard Ivry as the Senior Editor. The reviewers have opted to remain anonymous.

The reviewers have discussed the reviews with one another and the Reviewing Editor has drafted this decision to help you prepare a revised submission.

Summary:

This study uses eye movements collected from a large sample of human subjects (N=74) to test whether generalization of learned fear responses is driven by unspecific perceptual similarity or by perceptual similarity exclusively along dimensions that are relevant for distinguishing the CS+ from the CS-. The experimental design used 8 face stimuli which varied along two dimensions (gender and identity), spanning a circle. Subjects first viewed all 8 faces in a baseline session and then two faces (180 degree apart, thus varying only on one dimension) were associated with shock (CS+) and no-shock (CS-) in a conditioning session. Finally, subjects viewed all stimuli again in a generalization session. Fixation patterns in the baseline session were well predicted by a model of the circular organization of the stimulus set. However, pre-post conditioning changes were best explained by a model that separated the relevant and irrelevant dimension, and showed that changes occurred exclusively along the relevant stimulus dimension. This shows that aversive learning leads to specific changes only along the relevant stimulus dimension.

Overall the reviewers felt that this is an interesting study that uses a sophisticated analysis to address an important question. However, reviewers also had major concerns about the framing of the paper, the data and conclusions. Perhaps the most critical is the focus on generalization with limited ability to really test the two models of generalization using the current design, and the lack of evidence for the reproducibility of the findings.

Essential revisions:

1) The reviewers agreed that there is an element of this study which is potentially informative for models of generalization, but that it doesn't reduce to the simple competing/alternative theories set out by the authors because the study design is not optimal to test the hypothesis as stated. However, there was broad consensus among reviewers that the strongest and most interesting part of this paper is that fear conditioning directs eye movements to features of faces that are most relevant for predicting whether a shock will occur. This is related to the concept of active sensing (i.e., information-seeking) and conditioning driving attentional resources in the domain of face processing. This was the focus of the 2017 biorxiv paper (doi: doi.org/10.1101/125682), which concluded: "These findings show that aversive learning can introduce substantial remodeling of exploration patterns in an adaptive manner during viewing of faces." This conclusion is much closer to and fully supported by the data. The reviewers would therefore encourage the authors to rewrite their manuscript (and change the title) such that it focuses on the core findings concerning the effect of conditioning on exploratory eye movements, and fully move potential implications for models of generalization to the Discussion.

2) With the new focus on the basic patterns of eye movements, it would be important to also include more descriptive data to show what features in the fixation patterns drive the dissimilarity between faces along the relevant and irrelevant dimension. For instance, fixation heatmaps, scan paths, saccade statistics, blink statistics, etc. might be useful.

3) Reviewers were concerned that the current design does not dissociate effects of fear conditioning from the effects of repeated exposure to the CS faces during conditioning. This could be ruled out by separately analyzing the first and second half of the generalization session, with the prediction that if the effects are driven by differential exposure, the effect of the unspecific component should increase over time (i.e., after subjects are also exposed to all other stimuli). In contrast, no changes in the unspecific component would suggest that the results are driven by aversive learning rather than exposure per se.

4) Although the study sample is large (N=74), there were concerns that because the results come from a complex analysis, they may not replicate out-of-sample. Ideally, the authors would collect new data and show that the effects replicate in an independent sample. Alternatively, they could use analytical approaches (k-fold cross-validation, etc.) to provide more evidence that the results are reliable and reproducible. In particular, it would be important to show that the pre-post conditioning difference in CS+ and in CS- gaze patterns, and their interaction, is robust and replicable.

[Editors' note: further revisions were requested prior to acceptance, as described below.]

Thank you for resubmitting your work entitled "Fixation-Pattern Similarity Analysis Reveals Adaptive Changes in Face-Viewing Strategies Following Aversive Learning" for further consideration at *eLife*. Your revised article has been favorably evaluated by Richard Ivry as the Senior Editor, and three reviewers, one of whom is a member of our Board of Reviewing Editors.

The manuscript has been improved but there are some remaining issues that need to be addressed before acceptance, as outlined below:

Whereas reviewer 1 and 2 felt that the revised manuscript adequately addressed the initial concerns, reviewer 3 had some remaining comments. Based on our discussion, we ask:

1) The reviewers agreed that the manuscript would benefit from discussing two key limitations. These include an acknowledgement of the exploratory and relatively weak nature of the findings, and a discussion of possible disadvantages that come with the circular stimulus organization.

2) An additional concern regards the lack of an out-of-sample replication, which is compounded by the fact that only 65% of subjects behave according to the adversity gradient model. The reviewers are aware that collection of new data is not possible at this point. Instead, reviewer 3 recommends an additional analysis treating "model identity" as a random effect (e.g. using spm_BMS). In the event that the model comparison does not yield conclusive results, it should be discussed why this is the case. There are two possibilities why the group-level winning model only wins in 65% of the subjects. First, there is heterogeneity between subjects in the true mechanism. In this case, RFX is appropriate and should be used and reported. Or, the theoretical assumption is that they all use the same mechanism but data are too noisy for single-subject inference. It would be important if the authors could take a strong stance here (either model heterogeneity or noisy data), based on some theoretical considerations, and suggest appropriate future research steps to verify or falsify these conclusions.

---

## [Author Response]

Essential revisions:1) The reviewers agreed that there is an element of this study which is potentially informative for models of generalization, but that it doesn't reduce to the simple competing/alternative theories set out by the authors because the study design is not optimal to test the hypothesis as stated. However, there was broad consensus among reviewers that the strongest and most interesting part of this paper is that fear conditioning directs eye movements to features of faces that are most relevant for predicting whether a shock will occur. This is related to the concept of active sensing (i.e., information-seeking) and conditioning driving attentional resources in the domain of face processing. This was the focus of the 2017 biorxiv paper (doi: doi.org/10.1101/125682), which concluded: "These findings show that aversive learning can introduce substantial remodeling of exploration patterns in an adaptive manner during viewing of faces." This conclusion is much closer to and fully supported by the data. The reviewers would therefore encourage the authors to rewrite their manuscript (and change the title) such that it focuses on the core findings concerning the effect of conditioning on exploratory eye movements, and fully move potential implications for models of generalization to the discussion.

We thank the reviewers for underlining the strong points of the paper and guiding its revision into a more descriptive and data-based direction. We revised the Introduction so that models are presented more descriptively with regards to how aversive learning can change the geometrical structure of fixation pattern similarities. We now formulated the models and hypotheses in a way that touches generalization, but does not overstate potential implications. We interpret our models as possible shapes of similarity relationships, and not as models of fear generalization. Accordingly we changed names from Baseline and Perceptual Similarity to Perceptual Baseline and *Perceptual Expansion* for Model 01, from Adversity Categorization to *Adversity Gradient* for Model 02, and Adversity Tuning to *CS+ Attraction* for Model 03 (see Figure 1). References to the debate of perceptual versus threat-prediction accounts of fear generalization were moved to the Discussion (second paragraph).

Moreover, we changed the title back to “Fixation-Pattern Similarity Analysis Reveals Adaptive Changes in Face-Viewing Strategies Following Aversive Learning”.

We hope to have revised the manuscript such that changes in exploration patterns are presented more descriptively, and discuss implications for fear generalization in the discussion.

2) With the new focus on the basic patterns of eye movements, it would be important to also include more descriptive data to show what features in the fixation patterns drive the dissimilarity between faces along the relevant and irrelevant dimension. For instance, fixation heatmaps, scan paths, saccade statistics, blink statistics, etc. might be useful.

We support the notion to present a variety of descriptive data on patterns of eye movements, so that it becomes clear what changes in how humans explore faces when they become associated with potential harm. While our modelling approach based on individual fixation patterns was motivated by strong idiosyncrasies in scanning paths (e.g. Mehoudar, Arizpe, Baker, and Yovel, 2014), we agree with the reviewers that the found anisotropy does not answer explicitly, how exploration strategies change with learning.

We added new analyses and text on what drives the increased dissimilarity, i.e. discrimination of CS+ vs. CS- trials (see subsection “Spatial changes in exploration strategy”). To this end, we complemented the linear model with a machine learning (support vector machine, SVM) approach, trained SVMs on individual subjects to explore if one can decode CS+ from CS- trials, and which information is used for this classification process (see new Figure 4). We show that decoding accuracy is better along the specific dimension than for the orthogonal dimension, and improved from baseline to test phase only in the specific dimension. This mirrors the results from our linear models, thus corroborates our findings. To shed light on what differs between CS+ and CS- trials, we visualize activation maps of this classification (see Figure 4C). This way, we show which regions are most informative to decode along the specific dimension.

As our method of FPSA is based on spatial features of eye-movements (fixation density maps), other features as proposed by the reviewers (such as saccade statistics), were not considered in our model so far. Therefore, while cautious not to overly extend explorative analyses, we agree that it might be of interest to consider learning induced changes in these features as well, as they are common outcome measures in the literature on scanning behavior. We chose features that go beyond the spatial domain, and focused on traditional outcome measures, such as number and duration of fixations, saccade length and the entropy of FDMs as a compact summary statistic. The results are described in the subsection “Comparison of FPSA to common eye-movement features and ROI-based analyses”, and a plot on changes in these univariate outcome measures was added as Figure 5—figure supplement 2.

Moreover, we tried to link these features to individuals’ anisotropy obtained from FPSA, however none of these features could predict the anisotropy significantly (see the aforementioned subsection).

3) Reviewers were concerned that the current design does not dissociate effects of fear conditioning from the effects of repeated exposure to the CS faces during conditioning. This could be ruled out by separately analyzing the first and second half of the generalization session, with the prediction that if the effects are driven by differential exposure, the effect of the unspecific component should increase over time (i.e., after subjects are also exposed to all other stimuli). In contrast, no changes in the unspecific component would suggest that the results are driven by aversive learning rather than exposure per se.

We agree with the reviewers that our conditioning plan introduces exposure differences by showing only the CS+ and CS- face during the conditioning phase. This motivated the correlation of the individual anisotropy parameter with behavioral outcomes (Rating, SCR). The scatterplot of this relationship was added as Figure 3—figure supplement 1.

Going beyond this, we took up the suggestion to check for an increase in the unspecific component over time to rule out this potential bias. We analyzed the three runs separately and found that while the specific component increases from baseline to the first generalization phase run, subsequently staying on a stable level throughout the 2nd and 3rd run, the unspecific component does not increase with further exposure to these faces. We added a figure showing specific and unspecific components for each generalization run separately (Figure 3—figure supplement 2), as well as address this in the Results section (subsection “Multivariate fear tuning profiles in eye movements”, last paragraph).

4) Although the study sample is large (N=74), there were concerns that because the results come from a complex analysis, they may not replicate out-of-sample. Ideally, the authors would collect new data and show that the effects replicate in an independent sample. Alternatively, they could use analytical approaches (k-fold cross-validation, etc.) to provide more evidence that the results are reliable and reproducible. In particular, it would be important to show that the pre-post conditioning difference in CS+ and in CS- gaze patterns, and their interaction, is robust and replicable.

We agree with the reviewers about the complexity of the models. As argued in the manuscript, we chose this approach based on the strong idiosyncrasy in individual eye movements. However, this rightfully poses the question of how replicable the findings of the current 74 individuals would be in a new sample. While we strongly agree with the reviewers’ suggestion to see replication of the results in new samples, at present, we do unfortunately not have the resources to collect new data.

We therefore aimed to frame our findings more cautiously (e.g. effect size on individual level, subsection “Multivariate fear tuning profiles in eye movements”, fourth paragraph). We now additionally report the prevalence in the sample (65%) to indicate that not all subjects might show the effect. Yet, we think that the finding that the amount of anisotropy predicts behavioral outcome measures (ratings, SCR) does speak for the effect being induced by aversive learning and against overfitting within a large sample. Moreover, we complemented our FPSA approach with a second method, the SVM showing the same anisotropy, i.e. better classification accuracy along the specific dimension as compared to the orthogonal dimension.

Of course, a formal replication with new subjects would be preferable and we are hoping that our sharing the experimental script as well as code used for analysis openly encourages fruitful replication from other sites in the future.

[Editors' note: further revisions were requested prior to acceptance, as described below.]

The manuscript has been improved but there are some remaining issues that need to be addressed before acceptance, as outlined below:Whereas reviewer 1 and 2 felt that the revised manuscript adequately addressed the initial concerns, reviewer 3 had some remaining comments. Based on our discussion, we ask:1) The reviewers agreed that the manuscript would benefit from discussing two key limitations. These include an acknowledgement of the exploratory and relatively weak nature of the findings, and a discussion of possible disadvantages that come with the circular stimulus organization.

We thank the reviewers for this suggestion aimed at guiding the reader to a better understanding of possible limitations of the study. Concerning the choice of a circular stimulus continuum, we added a paragraph discussing the choice of our stimulus set to the Discussion section of the manuscript (sixth paragraph). Moreover, we discuss the need for replication and clarification more elaborately (Discussion, seventh paragraph), in line with the considerations about model heterogeneity or noise as underlying factors for the prevalence of 65%, as discussed below. We hope this contributes to describing the results of the present study in a transparent way including implications for and the necessity of further research.

2) An additional concern regards the lack of an out-of-sample replication, which is compounded by the fact that only 65% of subjects behave according to the adversity gradient model. The reviewers are aware that collection of new data is not possible at this point. Instead, reviewer 3 recommends an additional analysis treating "model identity" as a random effect (e.g. using spm_BMS). In the event that the model comparison does not yield conclusive results, it should be discussed why this is the case. There are two possibilities why the group-level winning model only wins in 65% of the subjects. First, there is heterogeneity between subjects in the true mechanism. In this case, RFX is appropriate and should be used and reported. Or, the theoretical assumption is that they all use the same mechanism but data are too noisy for single-subject inference. It would be important if the authors could take a strong stance here (either model heterogeneity or noisy data), based on some theoretical considerations, and suggest appropriate future research steps to verify or falsify these conclusions.

We have addressed this issue by performing an additional Gaussian Mixture Models (GMM) analysis, with which we determine whether the distribution of effect sizes of our effect of interest ((w_specific_gen_-w_unspecific_gen_) – (w_specific_base_-w_unspecific_base_)) is best explained by a model with a single Gaussian component or by models containing multiple Gaussian components (2-5 components). If the 65% fit on the single-subject level results from population heterogeneity, then one would expect one of the multiple Gaussian component models to explain the observed effect size distribution better than the single component model, and vice versa for the alternative explanation in line with a single mechanism with high noise. The results of this analysis reveal that none of the multiple component models outperformed the single component model (Figure 3—figure supplement 3). While the two component model’s BIC differed only slightly from the one component model (BIC_GMM1_ = -23.7 BIC_GMM2_ = -22.9, ΔBIC = -.8), its bigger complexity resulted in a second component only capturing outliers at the lower end of the distribution with a mixing proportion of only 4% (Figure 3—figure supplement 3B). We conclude that we do not find enough evidence for more than one subpopulation underlying the distribution of anisotropy values. The results of the analysis are mentioned in the Results section: (subsection “Multivariate fear tuning profiles in eye movements”, fifth paragraph) and in more detail in Figure 3—figure supplement 3 and its legend.

We consider the present study as a first step in exploring how fixation pattern similarities change after aversive learning. We tested our models on a continuum of differently strong changes, thus it is intuitive that these models capture the data differently well on the subject level. Therefore, in a next step, one would need to test predictions arising from our work in a more complex scenario allowing stronger predictions and therefore stronger evidence for or against specific models. One possibly already introduced in the Discussion paragraph of the manuscript would be to set up a more complex experiment with circles of different diameters. This enriched stimulus space would offer testing more complex predictions about dissimilarity relationships and this way would allow to replicate, verify or falsify our recent findings. As mentioned in the response to the first comment, we added these elaborations to the Discussion (Discussion, seventh paragraph), in line with the reviewers’ suggestion to discuss limitations and the small effect sizes further to guide the reader to a better understanding of the results.